

# Stress responses to conspecific visual cues of predation risk in zebrafish

Thiago Acosta Oliveira[1,*], Renan Idalencio[1,*], Fabiana Kalichak[1,*], João Gabriel dos Santos Rosa[1,*], Gessi Koakoski[1,*], Murilo Sander de Abreu[1,*], Ana Cristina Varrone Giacomini[1,2,*], Darlan Gusso[3,*], Denis Brook Rosemberg[1,*], Rodrigo Egydio Barreto[4,*] and Leonardo José Gil Barcellos[1,5,*]

[1] Programa de Pós-Graduação em Farmacologia, Universidade Federal de Santa Maria, Santa Maria, Brazil
[2] Bioscience Institute, Universidade de Passo Fundo, Passo Fundo, Brazil
[3] Department of Biologycal Sciences, Pontifícia Universidade Católica do Rio Grande do Sul, Porto Alegre, Brazil
[4] Department of Bioscience, Universidade Estadual Paulista, Botucatu, Brazil
[5] Programa de Pós-Graduação em Bioexperimentação, Universidade de Passo Fundo, Passo Fundo, Brazil
[*] These authors contributed equally to this work.

## ABSTRACT

Chemical communication relating to predation risk is a trait common among fish species. Prey fish under threat of predation can signal risk to conspecific fish, which then exhibit defensive responses. Fish also assess predation risk by visual cues and change their behavior accordingly. Here, we explored whether these behavioral changes act as visual alarm signals to conspecific fish that are not initially under risk. We show that shoals of zebrafish (*Danio rerio*) visually exposed to a predator display antipredator behaviors. In addition, these defensive maneuvers trigger antipredator reactions in conspecifics and, concomitantly, stimulate the hypothalamus-pituitary-interrenal axis, leading to cortisol increase. Thus, we conclude that zebrafish defensive behaviors act as visual alarm cues that induce antipredator and stress response in conspecific fish.

## INTRODUCTION

Prey–predator interactions occur throughout the animal kingdom (*Cresswell, 2010*), with every interaction involving both unique and general characteristics. Such interactions occur in multiple directions (prey to predator, predator to prey, prey to prey, and even predator to predator) (*Barcellos et al., 2014*; *Mullan, Glass & McCartney, 2015*; *Dunlop-Hayden & Rehage, 2011*). Prey can access predators through a diversity of signals that can be visual, olfactory, acoustical, and vibratory (*Barcellos et al., 2014*; *Barreto, Luchiari & Marcondes, 2003*). There is a vast amount of information in the scientific literature about these types of perceptions in fish (*Wisendem, Vollbrechet & Brown, 2004*; *Barcellos et al., 2011*) in response to the presence of a predator, and in conspecific fish threatened by a predator (*Jordão & Volpato, 2000*). Different combinations of these signals in the presence of a predator, or even the diverse forms of communication about a predator threat, between conspecific prey induce different antipredator maneuvers (*O'Connor et al., 2015*).

Corresponding author
Leonardo José Gil Barcellos,
lbarcellos@upf.br

During prey–predator interactions, early detection of predation risk is crucial for prey survival (*Allen, 1975*), and chemical communication plays a key role in such risk assessments (*Chivers & Smith, 1998*). Prey fish display antipredator behaviors when they perceive the odor of a predator (direct perception; e.g., *Miyai et al., 2011*) or when alerted to the presence of a predator via chemical cues released by other individuals (indirect perception; e.g., *Barcellos et al., 2011*; *Barcellos et al., 2014*; *Barreto et al., 2013*; *Oliveira et al., 2014*). Although an enormous amount of research has been undertaken on chemical communication among prey, many questions remain unanswered. In addition to chemical cues, prey fish also detect predators via visual cues and display appropriate defensive reactions in response to these threats (*Kalluef, Stewart & Gerlai, 2014*), but these responses are far less explored in the literature.

The visual presence of a predator induces changes in prey behavior and physiology (*Barreto, Luchiari & Marcondes, 2003*; *Gerlai, 2003*; *Barcellos et al., 2007*; *Miller & Gerlai, 2007*; *Gebauer et al., 2011*). As a diurnal fish species, zebrafish (*Danio rerio*) clearly use this sensorial modality for communication with conspecifics. Eavesdropping is one type of visual communication, wherein individuals use the available visual information provided by conspecific fish behaviors (*Abril-de Abreu, Cruz & Oliveira, 2015*) to evaluate the context and appropriate response behavior. It has been previously shown that eavesdropping influences zebrafish behavior (*Abril-de Abreu, Cruz & Oliveira, 2015*).

In this context, we propose that predator-induced behavioral changes could act as visual alarm signals that provoke defensive reactions in conspecific fish. Here, we show that antipredator behaviors displayed by zebrafish are visual alarm cues that induce defensive maneuvers and surges in the production of cortisol in unthreatened conspecifics.

## MATERIALS AND METHODS

### Ethical note

This study was approved (protocol #7/2013) by the Ethics Commission for Animal Use of the Universidade de Passo Fundo (Passo Fundo, RS, Brazil), and all methods were carried out in accordance with the guidelines of National Council of Animal Experimentation Control (CONCEA).

### Zebrafish and housing conditions

Wild-type zebrafish adults ($\pm 8$ months) of both sexes, averaging $\pm 5$ cm in length and $\pm 0.4$ g in weight, were maintained under a photoperiod of $\sim 14$ h light/10 h dark in indoor holding tanks (2 fish/L). Water conditions were maintained as follows: temperature: $28.0 \pm 2.0\,°C$; pH: $7.0 \pm 0.6$; dissolved oxygen: $6.8 \pm 0.4$ mg/L; total ammonia: $<0.01$ mg/L; total hardness: 6 mg/L; and alkalinity: 22 mg/L of $CaCO_3$. The fish were fed twice daily (09:00 h and 16:00 h) with commercial flakes (TetraMin®, Tetra, Melle, Germany) until satiation. The dimensions of the aquaria were 13 cm length $\times$ 30 cm width $\times$ 40 cm height, and contained a total water volume of 13 L; stock density was similar to the holding tanks (2 fish/L). The experimental chambers measured 40 cm length $\times$ 30 cm width $\times$ 30 cm height, and contained 30 L of water; stock density was 1 fish/3 L.

Each experiment was performed with four replicates and consisted of six groups containing 10 fish each (for a total of 240 fish). Additional controls were performed in triplicate, using two groups of 10 fish each (for a total of 60 fish). Thus, 300 fish were used in the experiments. For cortisol measurements, two data points were generated from each replicate (2–3 fish per data point) and then pooled into a single group, which was used for the statistical analysis.

## Experimental design and procedures

Zebrafish shoals were maintained either in a chamber that allowed visualization of a predator/non-predator fish or in a tank devoid of other fish. We called fish kept under these conditions "sender fish" (SF, Fig. 1). In an adjacent chamber, we placed other zebrafish shoals were placed in an adjacent chamber that only permitted visualization of the SF shoal. Fish kept under these conditions were termed "receiver fish" (RF, Fig. 1).

The chambers were tightly sealed to avoid potential chemical communication between fish in different chambers. Three experimental conditions were thus established: in the first, RF were evaluated during SF visual exposure to a predatory fish (the tiger oscar *Astronotus ocellatus*); in the second, a harmless fish (the goldfish *Carassius auratus*) was used to test whether visualization of any fish provoked behavioral changes in SF and, consequently, in RF. Lastly, SF were exposed to the view of a water-filled aquarium devoid of fish, which served as the control.

Two additional control conditions were established (Fig. 1) to ensure that RF could not view the predator directly. Two groups of RF were isolated (without the visual cues emanating from conspecific fish), one group remaining in their chamber 3 while the predator fish (tiger oscar) remained in its chamber 1, and a second group of RF was kept in their chamber 3 while the non-predator fish (goldfish) remained in its respective chamber 1. Sample size was $n = 10$ for each condition. Initially, SF and RF zebrafish shoals and stimulus fish (non-predator and predator) were introduced into their respective chambers for an adaptation period of 24 h, during which SF fish could not see the stimulus chamber (predator fish, harmless fish, or tank with only water), whereas RF were allowed to view SF during the entirety of the adaptation and experimental periods. Visual contact was blocked using an opaque plastic plaque. Following the adaptation period, the plaque was removed, thereby allowing visualization between the stimulus chamber and SF for 60 min, which was recorded on video for later behavioral analysis. After the visualization period, both SF and RF fish were captured, killed by immediate spinal section, frozen in liquid nitrogen, and stored at −20 °C until required for use in whole-body cortisol assays.

Chambers were cleaned prior to each replicate, but the aquaria were not cleaned, the water was not changed, and the fish were not fed during the experiment itself to avoid the effects of handling procedures, as fish remained in the chambers for only 24 h. No exchange of water occurred between the chambers; prior to the experimental trials, the chambers were filled with water and it was observed that adjacent chambers remained completely dry, thus ensuring that communication between fish within each chamber occurred solely via visualization.

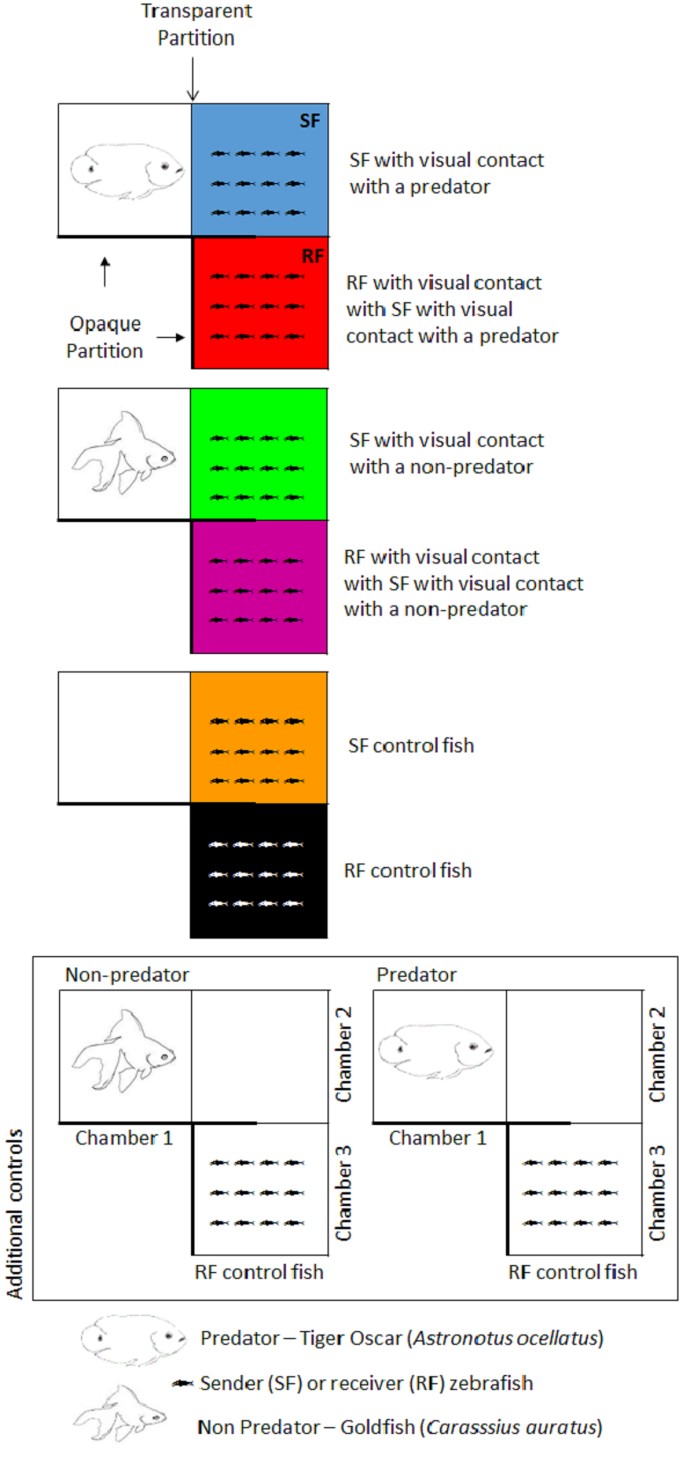

**Figure 1** **Schematic representation of the experimental conditions.** Fish illustrations were drawn by LB.

## Cortisol extraction and determination

Whole-body cortisol levels were determined in 4–6 fish specimens taken from each experimental group of each replicate, for a total of 16–24 fish. Cortisol sample sizes varied because whole-body cortisol analysis requires a minimum of 0.5 g of tissue extract; given the small size of these zebrafish, we used pooled samples of 2–3 fish, with the goal of attaining a final sample size of eight.

Tissue cortisol levels were used as an indicator of stress response. To minimize potential handling-induced stress responses, the period between capture and killing was <30 s. Each killed fish was weighed, minced, and placed in a tube containing 3 ml of phosphate buffered saline (PBSg, pH 7.3). The contents were then transferred to a tube containing ethyl ether and subjected to vortexing, then centrifuged after immediately being frozen in liquid nitrogen. The thawed portion (cortisol containing diethyl ether) was decanted and transferred to a new tube and allowed to evaporate completely to yield a lipid extract containing cortisol, which was then stored at $-20\,°C$.

Tissue extracts were re-suspended in PBSg and whole-body cortisol levels were measured using an ELISA kit (EIAgen$^{TM}$ cortisol test; BioChem Immunosystems, Burlington, VT, USA). This kit was previously shown to be effective for analysis of zebrafish tissue extracts by *Sink, Lochmann & Fecteau (2007)*. Accuracy was tested by calculating cortisol recovery in samples spiked with known amounts of cortisol; precision was assessed by calculating the intra-assay coefficient of variation (CV) of 12 repeated assays in seven randomly chosen samples on the same plate; and reproducibility was evaluated by assaying the same samples on different plates and calculating the inter-assay CV.

To test for linearity and parallelism, serial dilutions of tissue extracts were performed in the buffer provided with the ELISA kit. We detected a strong positive correlation between the curves ($R^2 = 0.892$) and determined that the samples displayed low inter- and intra-assay CVs (7–10% and 5–9%, respectively).

## Behavioral quantification

We used the methodology described by *Oliveira et al. (2014)* to quantify the behavioral responses. Briefly, the water column was divided into three sections of equal size from the bottom to the surface. The time that fish spent in the bottom area was observed and manually recorded by a single experienced observer who was unaware of what group was being assessed, with the percentage of the session time for each of these behaviors later calculated. The rationale for quantifying fish behavior via observation and manual recording was based on the findings of *Speedie & Gerlai (2008)*, which clearly show that zebrafish responses to alarm substances can be reliably quantified by visual-manual recording as well as through computerized video-tracking methods. We characterized time spent near the tank bottom as an indicator of defensive reactions (*Gerlai & Csányi, 1990*; *Gerlai et al., 2000*; *Quadros et al., 2016*), with the duration of this behavior expressed as a percentage of the total observation session duration. Onset of the time at the tank bottom was considered to occur when at least three of the 10 zebrafish remained within the bottom-third section, in accordance with the methods described by *Speedie & Gerlai (2008)*.

## Statistical analysis

For whole-body cortisol and behavior values, we compared all treatments and also performed comparisons between SF zebrafish senders and between RF zebrafish under experimental conditions. Regarding cortisol values and time spent in the bottom section of the aquaria, we applied a two-way ANOVA, with type of stimulus fish and category (SF or RF) of zebrafish as independent factors, followed by Tukey's multiple comparison test to compare the means in each experiment. Significant differences were set at $P = 0.05$. To compare the SF and RF zebrafish according to the type of stimulus fish, we applied a one-way ANOVA, followed by Tukey's multiple comparison test or a Kruskal–Wallis test followed by Dunn's multiple comparison test, depending on if data passed the Kolmogorov–Smirnov and Bartlett tests. Significant differences were set at $P = 0.05$.

## RESULTS

We detected a strong interaction between the type of stimulus fish and the category (SF or RF) of zebrafish ($P = 0.0077$; $F_{2,39} = 5.527$). We found that visual perception of the predator increased whole-body cortisol in both SF and RF zebrafish compared to the SF control ($P = 0.0077$ (Fig. 2A); SF comparison $P = 0.0006$, $K = 14.92$ (Fig. 2B); RF comparison $P = 0.001$, $K = 13.74$ (Fig. 2C)). This effect did not occur for SF and RF in the non-predator fish treatment, in which increased cortisol was observed only in SF relative to the SF control. The tank devoid of fish (filled with only water) elicited no significant cortisol response in either SF or RF (Fig. 2A).

Only the type of stimulus fish had a significant effect on defensive behavior ($P < 0.0001$; $F_{2,18} = 96.09$); both SF and RF fish exposed to a predator fish spent more in the bottom section of the tank, whereas SF and RF fish exposed to a non-predator fish also spent more time in the bottom section but this response was less intense (statistically lower) than those SF and RF observed in predator exposure treatment, considering the comparison of both groups (non-predator and predator treatment) with the control group (non-stimulus fish) (Fig. 2D). The aquaria with only water induced no significant changes in defensive behaviors (SF comparison $P < 0.0001$, $F_{2,9} = 50.17$ (Fig. 2E); RF comparison $P < 0.0001$, $F_{5,18} = 46.63$ (Fig. 2F)). Zebrafish from the two additional controls had very low cortisol concentrations ($2.88 \pm 0.74$ and $3.63 \pm 0.59$ ng/g tissue), levels typically found in non-stimulated control fish (*Barcellos et al., 2007*; *Barcellos et al., 2014*).

## DISCUSSION

Here we demonstrate that zebrafish visually exposed to predators display antipredator behaviors that in turn act as visual alarm cues for conspecifics. Zebrafish unexposed to a predator exhibited defensive maneuvers when allowed to view conspecific individuals displaying defensive behaviors. In addition, these visual cues induced activation of the hypothalamus-pituitary-interrenal axis (stress response), culminating in increasing levels of whole-body cortisol. These endocrine and behavioral responses are likely associated with stress events, such as predation risk, as they are well documented by studies of biological interactions between prey and predators in several species (*Barton, 2002*; *Clinchy*

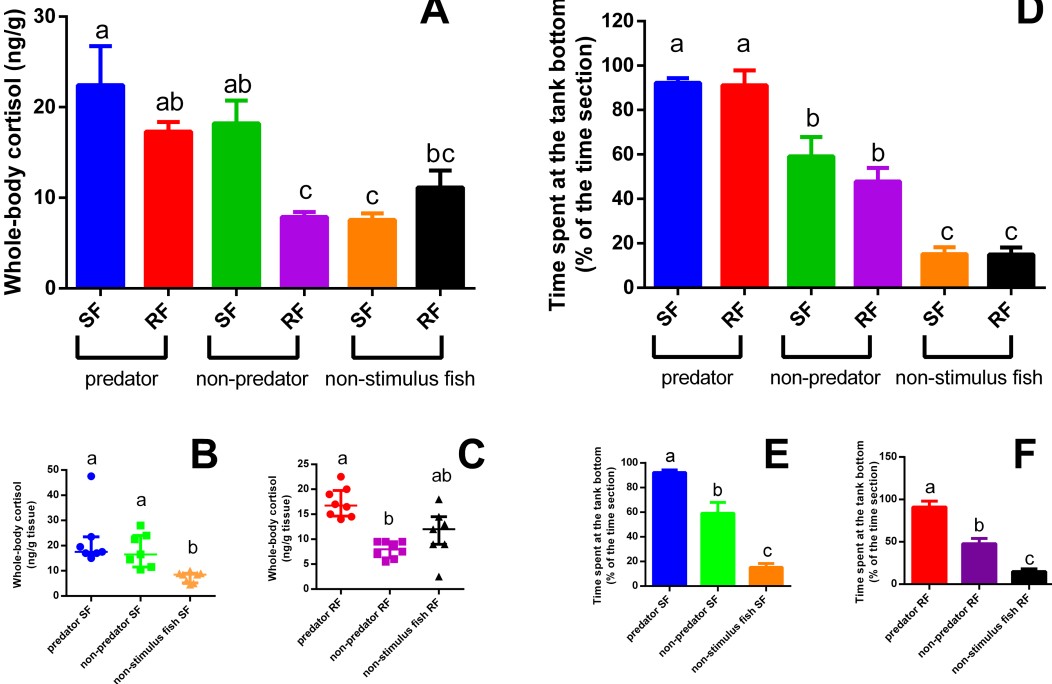

**Figure 2** **Whole-body cortisol response (mean + SEM) (A) and time spent in the tank bottom (mean ±SEM) (D) of SF (B and E) and RF (C and F).** Data of panels (A) and (D) were compared by two-way ANOVA followed by Tukey's multiple range test. Data of panels B and C were compared by Kruskal Wallis complemented by a Dunn's multiple comparisons test, while data of panels (E) and (F) by One-way ANOVA folowed by Tukey's multiple range test ($n = 7$–8 for cortisol and 4 for behavioral analysis). Different letters above medians or means indicate statistical differences.

*et al., 2011*; *Archard et al., 2012*). Our results suggest that visual eavesdropping occurs in zebrafish in the context of predation risk. Zebrafish in direct visual contact with a harmless fish, that is, a false predation risk, displayed short-term behavioral alterations and an irreversible stress response. We termed it "irreversible stress response" because we postulated that once triggered, this cortisol response displays a typical increase in cortisol levels in this group of fish, whereas behavioral responses can be quickly adjusted. In contrast, conspecifics observing this altered behavior were able to interpret and process the information as signaling a non-dangerous situation, evidenced by the lack of change in behavioral and endocrine stress responses in RF.

Cortisol levels were elevated in both SF and RF when a predator was visually accessible, presenting a condition of potential imminent predation risk, but cortisol increases were not observed in RF when in visual contact with a non-predator fish or a water-filled tank devoid of fish. Another reliable indicator of anti-predator behavior (*Gerlai, 2003*) is the time zebrafish spent at the bottom of the tank, which increased in SF when in visual contact with a predator as well as in their respective RF. We can confirm that zebrafish are capable of interpreting imminent predation risk simply by viewing the behavior of their conspecifics when a predator is present, which triggers an anti-predatory response. These results highlight the complexity of anti-predator communication among zebrafish, given

that defensive and stress responses can be triggered in conspecifics that cannot see the predator directly and thus represent indirect responses to predation risk. Communication based on visual cues aids zebrafish shoals in dealing with predators. Such communication might take the form of a chain reaction, in which a single fish detects a predator and responds to this threat with the defensive behavior consequently propagating throughout the entire shoal. Likewise, it may even be that zebrafish displaying anti-predator responses in one shoal can induce similar responses in fish in another shoal, given that copying is a commonly observed behavior in shoal fish, for instance when the leader of a shoal "commands" the group during foraging navigation (*Reebs, 2000*). Whether similar patterns occur for defensive reactions deserves future investigation.

That zebrafish in visual contact with a predator have higher levels of whole-body cortisol was previously reported by *Barcellos et al. (2007)*, but surprisingly, zebrafish within visual contact of a non-predator fish presented a similar increase in cortisol levels. However, the cortisol response within this group (SF non-predator treatment) was of smaller magnitude than in zebrafish viewing a predator. Nonetheless, RF viewing these conspecifics had no elevated cortisol. Cortisol levels in SF exposed to non-predator fish were also elevated, but these fish exhibited similar behavior (time in the bottom section of the tank). Thus, since RF received only visual cues, this behavior failed to elicit a cortisol response. In the non-predator treatment, both SF and RF spent more time near the tank bottom compared to fish in the control group (absence of a stimulus fish), but this response was not as significant as that among fish in the predator-exposure treatment.

The anti-predator response exhibited by fish in directly visualization of a predator (*i.e.*, SF) was a determinant factor for triggering anti-predatory behaviors in RF, confirming that communication of a potential threat from SF to RF occurred via visual cues. The unexpected behavioral and stress response in fish in the non-predator treatment may be due to limited assessment of the non-predator fish, given that zebrafish only had visual contact with the non-predator; recognition by fish of a real predation risk may depend on other cues, such as scent (*Korpi & Wisenden, 2001*) or mechanical (*Hegab & Wei, 2014*) cues. Visual recognition of a predator by prey fish is based on movement patterns characteristic of the predator (*Barcellos et al., 2007*); because our non-predator goldfish was very active, zebrafish may have initially mistaken it for a predator. Fish in our experiment were given only visual stimulus, and thus the absence of combinations of different cues may result in 'misinterpretation' of a harmless stimulus, leading to the observed increases in cortisol levels in SF in the non-predator treatment. Zebrafish that viewed conspecifics that in turn viewed a non-predator did not alter their behavior, and no cortisol response was triggered, supporting the 'misinterpretation' hypothesis. Our findings also highlight the need for care in relation to visual cues when using zebrafish in experiments. Other visual stimuli (e.g., presence of humans, other fish species, or even other types of animals) may be interpreted as potential threats by the fish, affecting their behavioral and hormonal patterns and consequently influencing experimental results.

The results of our study improve our understanding of predator–prey interactions and communication among conspecifics in response to predation risk in aquatic environments.

### Funding

This study was funded by the Universidade de Passo Fundo and CNPq (grant 470260/2013). L.J.G.B. holds CNPq research fellowships (301992/2014-2). The funders had no role in study design, data collection and analysis, decision to publish, or preparation of the manuscript.

### Grant Disclosures

The following grant information was disclosed by the authors:
Universidade de Passo Fundo and CNPq: 470260/2013, 301992/2014-2.

### Competing Interests

The authors declare there are no competing interests.

### Author Contributions

- Thiago Acosta Oliveira conceived and designed the experiments, performed the experiments, analyzed the data, wrote the paper.
- Renan Idalencio, Fabiana Kalichak, João Gabriel dos Santos Rosa, Murilo Sander de Abreu and Darlan Gusso performed the experiments.
- Gessi Koakoski and Ana Cristina Varrone Giacomini contributed reagents/materials/-analysis tools.
- Denis Brook Rosemberg analyzed the data, wrote the paper, reviewed drafts of the paper.
- Rodrigo Egydio Barreto conceived and designed the experiments, wrote the paper, reviewed drafts of the paper.
- Leonardo José Gil Barcellos conceived and designed the experiments, analyzed the data, wrote the paper, prepared figures and/or tables.

### Animal Ethics

The following information was supplied relating to ethical approvals (i.e., approving body and any reference numbers):

This study was approved by the Ethics Commission for Animal Use of Universidade de Passo Fundo (Passo Fundo, RS, Brazil), and all methods were carried out in accordance with the guidelines of National Council of Animal Experimentation Control (CONCEA).

### Data Availability

The raw data is included as Supplemental Information 1.

### Supplemental Information

Supplemental information for this article can be found online at http://dx.doi.org/10.7717/peerj.3739#supplemental-information.

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
