# Peer review of "Stress responses to conspecific visual cues of predation risk in zebrafish"

_PeerJ, doi:10.7717/peerj.3739_

## Round 0.1 · original submission · Major Revisions

· Academic Editor

Major Revisions

The reviewers prepared several comments and queries for your attention, and I have attached a copy of this report below. I believe that all comments will improve your manuscript quality. I suggest responding all queries point-by-point.

Reviewer 1 ·

Basic reporting

Although the manuscript has been proof read by a native English speaking coleague, I do feel the document could use another proof reading to correct some sentence structures and remove grammatical errors. For example line 46-48 could use a rewrite. In addition, I am not sure if the word ‘subjugated’ is the proper term to use in this context. I strongly suggest the authors have a second native English speaker read and correct the document.

The manuscript seems to contain sufficient background information and is well supported with literature references.

The manuscript is conform the correct layout of the journal (Single File, Figures embedded in text). The authors also supplied the raw data and statistics.

The manuscript holds a plausible hypothesis, as the authors wish to explore to presence of visual eavesdropping when it comes to predator-induced behaviour. However, the authors suggest this information could be used in experiments towards Autism Spectrum Disorder for example. I would have liked to see a paragraph in the discussion that links back to how their findings would fit into such experiments (or if more experiments are needed to do so).

Experimental design

The research provided in the manuscript is original and fits within the scope of the journal.

The authors have clearly stated a research question in their Introduction. However, the relevance of their study feels lacking. There is mention of use in studies towards specific drugs and/or Autism Spectrum Disorder, but these examples are only mentioned. I would like to see how their study (visual eavesdropping of predatory-induced behaviour causes increased cortisol levels and predatory-induced behaviour in observer fish) fits into (future) studies on the mentioned topics.

The experiment appears to be conducted in accordance with local ethical laws towards the handling of animals.

However, the experiment seems lacking in the technical area. Whether this is due to insufficient elaboration on the methods, is unclear. I will therefore state my concerns with the experimental approach.
[1] I understand that all experiments were conducted in triplicate (based upon the total number of fish used and the number of fish per group). Please state this clearly in the Material and Methods section
[2] I am confused on how cortisol values were determined. The text states:
“In each group, 10 fish were used at each sampling time, forming 5 pooled samples of 2 fish each.”
What sampling times are the authors referring too?
I understood that each group was only 10 fish in size, thus all fish in a group were used for cortisol analysis?
Why did the authors pool the material of two fish into a single sample point?
How many sample points are there in total per group? (5 sample points per replicate, 3 replicates = 15 data points?)
[3] Regarding the video analysis. Why did the authors not split their video analysis in different time periods to get a better indication on habituation to the ‘predator’ tank? The authors did mention in their discussion, that the data on the behavioural response to the gold fish might have been ‘polluted’ by an initial ‘false’ predator-induced behavioural response in the first 5 minutes. I strongly suggest re-analysing the data (since it was recorded) to show behavioural changes over time.

Validity of the findings

The study seems novel in its findings and adds to the previous reports of eavesdropping in zebrafish.

I have a few major concerns with the statistics in this study and I would like the authors to elaborate more on their choice of statistical analysis.
[1] Please include the results for ANOVA and/or Kruskal Wallis tests in the figures (or legends: p-value, df, F-value). Also include the total number of data points per group.

[2] Why did the authors chose to display S.E.M. and not S.D. in the graphs? And have the authors considered presenting their data as individual dots or box-plots (instead of a bar graph). This would provide a lot more information on the spread and distribution of the samples.

[3] Seeing how one of the analysis required a Kruskal Wallis, were there signs of two groups within the population (see point 2 on individual data points)

[4] Why did the authors analyse their data with an ANOVA/Kruskal Wallis test? Would other statistical approaches not be better suited to analyse the data-sets? By using an ANOVA, the authors also compare groups that have no need for comparison (for example: the RF fish in the Empty Tank experiment, compares to SF fish in the Predator Experiment). By adding these comparisons, the power of the statistical test goes down. In addition, when comparing all groups like the authors have done, would a two-way ANOVA not be the correct analysis to use (there are two variables that are being compared: Group of Zebrafish and Contents of Experiment Tank).

I think the authors should ask themselves what it is they wish to show with their data and match the statistical analysis to that specific question. To me, it appears they wish to show the following:
Is there a change in behaviour and physiology in SF fish when exposed to a predator or a non-predator fish [Does our experimental setup work? -> analyse SF control, predator and non-predator]
Is there a change in behaviour and physiology in the RF fish when the SF fish are exposed to a predator or a non-predator [Do we find a change in behaviour in RF fish when our SF fish displayed changes? -> analyse RF fish control, predator and non-predator]
Is there a difference in behaviour and pysiology between SF fish and RF fish under the three conditions [Are the changes we find similar between SF and RF fish? -> analyse SF fish and RF fish under control, predator and non-predator conditions]

[5] I suggest changing the way statistical significance is displayed in the figures. As it is now, it is very confusing. Use letters to indicate significantly different groups (different letter). Use capital letters and lowercase letters to seperate groups within a single figure (for example ‘AAB’ for an ANOVA done across the RF fish and ‘abb’ for an ANOVA done across the SF fish).

The conclusions presented in the manuscript are not always backed up by the data. This may in part be due to the lack of clear research questions that are answered by the data (see previous statement on statistical analysis). In this regard, it would be good to split the discussion into sections (one section for each question answered by the data and statistical analysis). In its current form, I have some concerns with the following conclusions:
LINE 251-252. the authors state that RF fish in the non-predator experimental conditions had no changes in cortisol and behaviour. However, this appears to be only true for cortisol values? Their behaviour appears significantly different from the control group?

LINE 265-268: Split the video data? Analyse first 5 minutes and last 5 minutes to show changes in behaviour during the 1h exposure period? See previous comments on Experimentl Design
LINE 269-271: But RF fish DID change their behaviour (compared to controls). However, cortisol levels were not different.
LINE 265-271: The authors state that the changes in behaviour and cortisol might be due to ‘misinterpretation’ of the goldfish by the SF fish. The authors refer to other cues (such as smell and sound) that could have informed the zebrafish that the gold fish is not a predator and thus the absence of these cues may have caused the change in behaviour and physiology. However, it should be noted that the authors only analysed position within the tank (scoring time spent at the bottom) of the zebrafish. It is possible that the SF fish display other types of behaviour that informs the RF fish of the presence of another non-predatory fish. The lack of a cortisol response in the RF group suggests there is a difference in the behaviour of the SF predator and non-predator group. Perhaps examining the recordings may give more insight.

LINE 272-275: Please elaborate what is meant by ‘others visual stimulus’. Although true that their behaviour and physiology did change in response to a gold fish, there is no evidence that similar changes occur when exposed to ‘other visual stimiulus’ - this response may be specific for exposure to other fish species?

LINE 277-282: I agree with this conclusion. Experiments that alter group behaviour in fish should be done out of sight of other experimental groups of fish. These data indicate effects on behaviour and physiology as a result of eavesdropping.

Additional comments

The manuscript brings an interesting data-set that adds to the recent findings on eavesdropping in zebrafish.

However, I do feel that the authors have not gotten the most out of the results that they have found. Specifically the approach in analysing the behavioural data as well as performing statistical analysis on their data could provide more (and important) information that will strenghten the manuscript.

I have advice a major revision of the manuscript, as I feel some new data needs to be generated (fairly easy to do, as the raw material is available) and the analysis of the data needs to be expanded. In addition, the manuscript as a whole would become better when the finds from the data are split into seperate research questions.

I have provided my thoughts and suggestions on how to improve the manuscript in each of the section boxes above.

·

Basic reporting

There are a lot of typing, syntax and grammar errors, throughout the text. The authors are encouraged to get editing help from someone with full professional proficiency in English.

Introduction should be improved by focusing on the subject (i.e. general and specific cues for predator detection; visual cues and anti-predator behavior) and by adding relevant literature on the field (e.g. Wisenden et al., 2004. Animal Behavior, 67: 59-67; Holms & McCormic, 2010; Dunlop-Hayden & Rehage, 2011. Behaviour, 148:795-823; O’Conor et al., 2015. Behavioural Processes, 121: 21-29 etc.)

Experimental design

The submission is in line with the aims and scope of the Journal.

The research question is clearly defined and Figure 1 (schematic representation of the experimental conditions) is very useful to understand and evaluate the experimental procedure. It is recommended to add information on the size and sex of fish used, as well as on the dimensions of the experimental chambers – in relation to the stocking density (number of fish per L). The authors should explain why they include only one parameter (time spent near the tank bottom) for behavioral quantification of fear behavior and did not use other parameters too, like shoal cohesion, faster swimming with spontaneous rapid turns, increased frequency and length of freezing bouts etc. (see Egan et al., Behav. Brain Res., 2010).

The research has been conducted in conformity with the prevailing ethical standards in the field.

Validity of the findings

There is a major concern on the statistical analysis. The authors should apply logarithmic transformation of cortisol data to check for the normality criterion. In addition, multivariate analysis of variance (MANOVA) should be applied and/or Two-way ANOVA to quantify/specify the difference among the experimental groups [Factor 1: fish status (sender – receiver); Factor 2: Trial (Predator, Non-predator, Non stimulus). The authors should provide an explanation on why (a) receiver – non-predator exposed fish have statistically lower whole-body-cortisol concentrations than sender-non-predator fish), and (b) non-stimulus fish have quite higher cortisol concentrations from the basal values reported in other published studies on zebrafish. Finally, the authors should check the appropriateness of ANOVA for percentage comparisons of the behavioral parameter “time spent near the tank bottom) and to add date on the number of fish that were located near by the bottom of the tank per session.

Additional comments

The authors use the terms "sender" and "receiver" fish. Is this an established / common accepted term or just a neologism?

---

## Round 0.2 · Major Revisions

· Academic Editor

Major Revisions

I received the review comments regarding your last manuscript version. You will see that one reviewer has raised points that need to be addressed by a revision. Moreover, I raised some points to be addressed by the authors. I, therefore, invite you to revise and resubmit your manuscript, considering the points raised.

Let me reinforce some of the reviewer’s comments and add some more:

(1) Please keep Figure 2 consistent by using the same colors and symbols to denote the different groups.
(2) Please incorporate your response to the following issue raised by reviewer #2 in the manuscript: “The authors should provide an explanation on why (a) receiver – non-predator exposed fish have statistically lower whole-body-cortisol concentrations than sender-non-predator fish)”.
(3) Please check the text added in the discussion section, page 13, line 269, “However, this response was smaller when compared with the predator exposure treatment, being the response of the SF- nonpredator higher than his respective RF”. There is no statistical difference between non-predator SF and RF fish, so higher values cannot thus be reported as such.
(4) There are some typos in the text. Please revise.

Moreover, please check the file for the comments from reviewer #1.

Reviewer 1 ·

Basic reporting

see attachment

Experimental design

see attachment

Validity of the findings

see attachment

Additional comments

The authors have adequately addressed most of the comments and concerns raised previously. However, I still have some concerns and questions that I feel require addressing. Please see the attached document.

Annotated reviews are not available for download in order to protect the identity of reviewers who chose to remain anonymous.

---

## Round 0.3 · Minor Revisions

· Academic Editor

Minor Revisions

The reviewer made some final observations regarding your manuscript. I recommend you to revise the English and clarify the experimental details as requested by the referee. With these final improvements I will be able to accept your manuscript for publication.

Reviewer 1 ·

Basic reporting

n/a

Experimental design

The authors clarified the confusion on where data points were coming from. However, this is still not very clear in the text. It is also hard to figure out how often the additional controls are performed (total number of fish suggest triplicates).

Data Points:
Experiments were performed in 4-fold and from each replicate two data points were generated (2-3 fish per data point, see cortisol extraction). These data points were then pooled into a single group and used for statistical analysis.

Number of fish:
Each experiment was done in 4-fold and consisted of 6 groups, containing 10 fish each (240 fish total). In addition, we performed additional controls in triplicate?, using two groups of 10 fish each (for a total of 60 control fish). This means 300 fish were used for the experiment, however for cortisol extraction only 4-6 fish were sacrificed per group in each experiment for a total of xxx fish.

Validity of the findings

n/a

Additional comments

The authors should really be more clear and accurate in their reporting. A lot of confusion could have been avoided if they had written down their procedures with more detail/information and more clearly. Also, between reviews the total number of replicates and the number of fish changed, which may give doubt to the accuracy of their other reportings.

Lastly, I am still not sure what the authors try to show in the first (large) set of graphs and the ANOVA. Performing the ANOVA across that data-set is not wrong, but it offers limited (and in my view useless) information (what is the added value of comparing SF predator exposed fish to RF non-stimulus fish?)

From their rebuttal it was not clear why the ANOVA was performed (language barrier perhaps), but if the authors wish to show that ALL SF groups have an increased or decreased response compared to ALL RF groups, then they should perform a two-way ANOVA (as suggested in the first review session). Similarly, the two-way ANOVA will shows if Predator exposed fish (so RF and SF together) show a greater or smaller response compared to for example (non-predator RF and SF fish together). Such conclusions can not really be drawn from the statistical analsysis done now, as a one-way ANOVA does not correct for this type of analysis.

---

## Round 0.4 · accepted · Accept

· Academic Editor

Accept

I am pleased to inform you that your manuscript has been accepted for publication in PeerJ.